# Dominating lengthscales of zebrafish collective behaviour

Yushi Yang[1,2]*, Francesco Turci[2], Erika Kague[3], Chrissy L. Hammond[3], John Russo[4], C. Patrick Royall[2,5,6]

1 Bristol Centre for Functional Nanomaterials, University of Bristol, Bristol, United Kingdom, 2 H.H. Wills Physics Laboratory, University of Bristol, Bristol, United Kingdom, 3 Department of Physiology, Pharmacology, and Neuroscience, Medical Sciences, University of Bristol, Bristol, United Kingdom, 4 Department of Physics, Sapienza Università di Roma, Rome, Italy, 5 Gulliver UMR CNRS 7083, ESPCI Paris, Université PSL, Paris, France, 6 School of Chemistry, University of Bristol, Bristol, United Kingdom

* yushi.yang@bristol.ac.uk

## Abstract

Collective behaviour in living systems is observed across many scales, from bacteria to insects, to fish shoals. Zebrafish have emerged as a model system amenable to laboratory study. Here we report a three-dimensional study of the collective dynamics of fifty zebrafish. We observed the emergence of collective behaviour changing between ordered to randomised, upon adaptation to new environmental conditions. We quantify the spatial and temporal correlation functions of the fish and identify two length scales, the persistence length and the nearest neighbour distance, that capture the essence of the behavioural changes. The ratio of the two length scales correlates robustly with the polarisation of collective motion that we explain with a reductionist model of self–propelled particles with alignment interactions.

**Data Availability Statement:** All relevant data are within the manuscript and its Supporting information files.

**Funding:** CLH and EK were funded by Versus Arthritis grants 21937 and 21211. YY was funded

## Author summary

Groups of animals can display complex collective motion, which emerges from physical and social interactions amongst the individuals. A quantitative analysis of emergent collective behaviour in animals is often challenging, as it requires describing the movement of many individual animals. With an innovative 3D tracking system, we comprehensively characterized the motion of large groups of zebrafish (*Danio rerio*), a freshwater fish commonly used as a vertebrate model organism. We find that the different collective behaviours are captured by two physical scales: the length of persistent motion in a given direction and the typical nearest neighbour distance. Their ratio allows us to interpret the experimental results, in the light of a statistical mechanics model for swarming with persistent motion and local neighbourhood alignment.

## Introduction

In living systems aggregation occurs at different scales, ranging from bacteria (microns) to insects (centimetres) to fish shoals (tens of kilometres) and with emerging complex patterns

by the China Scholarship Council (grant 201700260078), and supported by a studentship provided by the Bristol Centre for Functional Nanomaterials (EPSRC Grant EP/L016648/1). The funders had no role in study design, data collection and analysis, decision to publish, or preparation of the manuscript.

**Competing interests:** The authors have declared that no competing interests exist.

[1–3]. These manifestations of collective behaviour originate from the interactions among the individual agents and between the agents and the environment [4]. Such interactions are often modelled by a combination of deterministic and stochastic contributions, capturing the individual's variability observed in nature and unknown or uncontrollable variables. The emergence of collective behaviour has been shown to be advantageous for communities [5–7], and the identification of universal patterns across scales and species reveals the physics behind these phenomena [8, 9]. Understanding the relationship between the collective behaviour and animal interactions has potential technological applications, for example to reverse engineer algorithms for the design of intelligent swarming systems [10]. Successful examples include the global optimisation algorithm for the travelling salesman problem inspired by the behaviour of ants, and implementation of the Boids flocking model in schooling of robotic fish [11, 12].

In a reductionist approach, collective behaviour can be modelled with interacting agents representing individuals in living systems. For example, groups of animals may be treated as if they were self-propelled particles with different interacting rules [13, 14]. Examples of using simple agent–based models applied to complex behaviour include describing the curvature of the fish trajectories as a Ornstein–Uhlenbeck process [15], modelling the ordered movement of bird flocks by an Ising spin model [2, 16], mapping of midge swarms onto particulate systems to explain the scale-free velocity correlations [13, 17, 18] and swarming in active colloids [19, 20]. One of the simplest approaches is the Vicsek model [21], in which the agents only interact via velocity alignment. Despite its simplicity, a dynamical phase transition from ordered flocking to randomised swarming can be identified, providing a basis to describe collective motion in biological systems [22, 23].

The study of collective behaviour in living systems typically has focused on two-dimensional cases for reason of simplicity, making the quantitative characterisation of three-dimensional systems such as flocks of birds or shoal of fish rare. To bridge this gap, zebrafish *(Danio rerio)* present a wealth of possibilities [24]: zebrafish manifest shoaling behaviour, i.e. they form groups and aggregates, both in nature and in the laboratory; also, it is easy to constrain the fish in controlled environments for long–time observations. Typically, the response of fish to different perturbations, such as food and illumination, can be pursued [24–26]. Furthermore, genetic modification has been very extensively developed for zebrafish, giving them altered cognitive or physical conditions, and yielding different collective behaviour [27, 28].

However, tracking zebrafish in three dimensions (3D) has proven difficult [29]. To the best of our knowledge, previous studies on the 3D locomotion of zebrafish focussed either on the development of the methodology [30, 31], or were limited to very small group sizes ($N \leq 5$) [29, 32, 33], while ideally one would like to study the 3D behaviour of a statistically significant number of individuals, representative of a typical community. In the field, zebrafish swim in 3D with group sizes ranging from tens to thousands [34].

Here we report on the collective behaviour of a large group ($N = 50$) of wild-type zebrafish, captured by a custom 3D tracking system. The observed fish shoals present different behaviours, showing different levels of local density and velocity synchronisation. We identify two well-separated time scales (re-orientation time and state-changing time) and two important length scales (persistence length and nearest neighbour distance) for the zebrafish movement. The time scales indicate the fish group change their collective state gradually and continuously. The spatial scales change significantly as collective behaviour evolves over time, with strong correlations between spatial correlations and shoaling. Finally, we reveal a simple and universal relationship between the global velocity alignment of the shoals (the *polarisation*) and the the ratio between the two length scales (the *reduced persistence length*). We rationalise this finding through the simulation of simple agent-based models, in which an extra inertia term is

added to the Vicsek model. Our findings illustrate complex behaviour in zebrafish shoaling, with couplings between spatial and orientational correlations that could only be revealed through a full three-dimensional analysis.

# Materials and methods

## Ethics statement

The experiments were approved by the local ethics committee (University of Bristol Animal Welfare and Ethical Review Body, AWERB) and given a UIN (university ethical approval identifier).

## Zebrafish husbandry

Wildtype zebrafish were kept in aquarium tanks with a fish density of about 5 fish / L. The fish were fed with commercial flake fish food (Tetra Min). The temperature of the water was maintained at 25˚C and the pH $\approx$ 7. They were fed three times a day and experience natural day to night circles, with a natural environment where the bottom of the tank is covered with soil, water plant, and decorations as standard conditions [35]. Our young group (Y) were adults between 4–6 months post-fertilisation, while our old group (O) were aged between 1–1.5 year. The standard body lengths of these fish were are available in section II in S1 Text. All the fish were bred at the fish facility of the University of Bristol.

## Apparatus

The movement of the zebrafish were filmed in a separate bowl-shaped tank, which is immersed in a larger water tank of 1.4 m diameter. The radius $r$ increasing with the height $z$ following $z = 0.734r^2$. The 3D geometry of the tank is measured experimentally by drawing markers on the surface of the tank, and 3D re-construct the positions of the markers. Outside the tank but inside the outer tank, heaters and filters were used to maintain the temperature and quality of the water. The videos of zebrafish were recorded with three synchronised cameras (Basler acA2040 um), pointing towards the tank. Detailed information is available in subsection I.A in S1 Text.

## Measurement and analysis

Fifty zebrafish were randomly collected from their living tank, moved to a temporary container, then transferred to the film tank. The filming started about 10 minutes after fish were transferred. The individual fish in each 2D images were located by our custom script and we calculated the 3D positions of each fish following conventional computer vision method [36, 37]. The 3D positions of the fish were linked into trajectories [38, 39]. Such linking process yielded the positions and velocities of different fish in different frames. We segmented the experimental data into different sections of 120 seconds, and treat each section as a steady state, where the time averaged behavioural quantities were calculated. More details on the tracking are available in subsections I.B-I.D in S1 Text, and more descriptions on the analysis are in section III in S1 Text.

# Results

## Experimental observation

We tracked the movement of zebrafish from multiple angles using three synchronised cameras. We collected data for fish groups with different ages, with young fish (labelled as Y1–Y4) and old fish (labelled as O1–O4). Fig 1A schematically illustrates the overall setup of the

experiment, where the cameras were mounted above the water to observe the fish in a white tank in the shape of a parabolic dish, enabling 3D tracking [2, 40–42]. With this apparatus, we extract the 3D positions of the centre of each fish at different time points, with the frequency of 15 Hz. We then link these positions into 3D trajectories. Fig 1B presents typical 3D trajectories from 50 young zebrafish during a period of 10 seconds, where the fish group changed its moving direction at the wall of the tank. The zebrafish always formed a single coherent group, without splitting into separate subgroups during our observations. Supplementary videos (S1, S2 and S3 Videos) are examples of the their movements. Fig 1C shows the cumulative spatial distribution of the zebrafish in the tank, during a one-hour observation. It is clear from this figure that the fish tend to swim near the central and bottom part of the tank. The propensity of zebrafish to swim near the wall was our motivation to use a bowl-shaped fish tank shown in Fig 1C, so that there are no corners for the fish to aggregate in, compared to a square-shaped container like conventional aquaria.

## Evolving collective behaviour

The 3D tracking yields the positions of the fish, whose discrete time derivative gives the velocities. From these two quantities, we calculate three global descriptors to characterise the behaviour of the fish: the average speed, the polarisation, and the nearest neighbour distance. The average speed is defined as $v_0 = 1/N\Sigma|\mathbf{v}_i|$ where $i$ runs over all the tracked individuals. The polarisation $\Phi$ characterises the alignment of the velocities. It is defined as the modulus of the average orientation, written as [13], $\Phi = 1/N|\Sigma(\mathbf{v}_i/|\mathbf{v}_i|)|$ where $i$ runs over all the individuals. Large polarisation ($\Phi \sim 1$) signifies synchronised and ordered movement, while low polarisation indicates decorrelated, random movement. The nearest neighbour distance between the fish centres is defined according to the Euclidean metric, and we focus on is arithmetic mean $l_{\text{nn}}$. These quantities were selected, because $v_0$ and $\Phi$ describe the dynamic of the fish, and $l_{\text{nn}}$ captures structural information on the group of fish.

We start from the analysis of temporal correlations of these three scalar quantities. Notably, all three exhibit two distinct time scales. Fig 2A shows the auto–correlation functions (ACF) of $v_0$, $\Phi$ and $l_{\text{nn}}$ averaged over the group of 50 young fish, calculated from a one hour observation. The ACFs present two decays and one intermediate plateau. We identify the first decay ($\sim 1$s) corresponding to the reorientation time of the zebrafish. This can be shown through the analysis of the autocorrelation of the orientations Fig 2B, which are characterised by an exponential decay with relaxation time $\langle\tau\rangle$ close to $\sim 1$s. This value is compatible with the previously reported turning rate timescale ($\sim 0.7$s) [43].

The plateau and subsequent decay of the ACF of the scalar quantities $v_0$, $\Phi$ and $l_{\text{nn}}$, with the time scale of $\sim 120$ seconds, represent complete decorrelation from the initial state, indicating that the shoal properties change significantly on this much longer timescale. Therefore, we employ time-windows of 120 seconds to average the time evolution of of $v_0$, $\Phi$ and $l_{\text{nn}}$, to characterise the states of the fish groups with moving averages $\langle v_0 \rangle(t)$, $\langle\Phi\rangle(t)$ and $\langle l_{\text{nn}}\rangle(t)$.

To characterise the degree of spatial correlation of the fish, we calculate their radial distribution function (RDF), see Fig 2C, which quantifies the amount of pair (fish-fish) correlations and it is commonly employed in the characterisation of disordered systems ranging from gas to liquids, from plasma to planetary nebulae [44]. Details on the RDF can be found in subsection III.C in S1 Text. All the RDFs exhibit one peak at a short separation, indicating the most likely short-distance separation between fish. The peak height is a measure of the cohesion of the fish group. Inspired by liquid state theory [44], we take the negative logarithm of the peak height to characterise what we call as the "effective attraction" among the fish, noted as $\langle\epsilon\rangle$. While $l_{\text{nn}}$ quantifies a characteristic lengthscale in the macroscopic collective state, $\epsilon$ quantifies

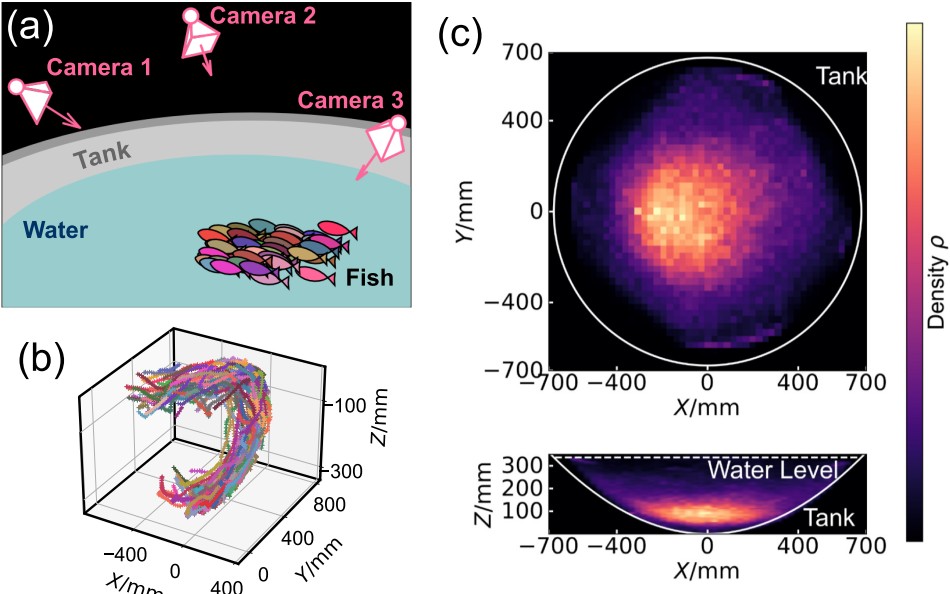

**Fig 1. Experimental setup and overall spatial distributions.** A: Schematic illustration of the experimental setup. Zebrafish were transferred into a bowl-shaped tank and three cameras were mounted above the air-water interface to record the trajectories of the fish. B: 3D trajectories obtained from the synchronised videos of different cameras. These trajectories belong to 50 young zebrafish (group Y1) in 10 seconds. C: The spatial distribution of 50 young fish (Y1) during a one-hour observation. Brighter colour indicates higher density. The top panel shows the result in XY plane, obtained from a max-projection of the full 3D distribution. The bottom panel shows a max-projection in the XZ plane. The outline of the tank and water-air interface, obtained from 3D measurement, are labelled.

the fish propensity to remain neighbours. In Fig 2D we see that $\langle l_{\mathrm{nn}} \rangle$ and $\langle \epsilon \rangle$ are strongly correlated, confirming that $l_{\mathrm{nn}}$ is also a measure of the cohesion of the collective states. We term $\langle v_0 \rangle$, $\langle \Phi \rangle$, $\langle \tau \rangle$, $\langle l_{\mathrm{nn}} \rangle$, and $\langle \epsilon \rangle$ "behavioural quantities", and the brackets indicate the moving average. These variables are summarised in Table 1.

Fig 2D illustrates the time-evolution of all the behavioural quantities, calculated from the movement of 50 young fish (group Y1) ten minutes after they were extracted from a husbandry aquarium and introduced into the observation tank. Over time, the behavioural quantities drift, indicating that a steady state cannot be defined over the timescale of 1 hour. This result is generic, as the separated time scales and changing states were obtained from repeated experiments on the fish group (Y2–Y4), and also from different groups of older zebrafish (O1–O4).

## Shoaling state described by two length scales

To describe the space of possible collective macroscopic states we employ two dimensioned lengths, the nearest neighbour distance $\langle l_{\mathrm{nn}} \rangle$ defined above and a second scale characterising the typical distance that a single fish covers without reorientation, the persistence length $\langle l_p \rangle$. This is defined as the product of the speed and the orientational relaxation time $\langle l_p \rangle = \langle v_0 \rangle \langle \tau \rangle$.

The resulting $\langle l_p \rangle$ and $\langle l_{\mathrm{nn}} \rangle$ diagram is illustrated in Fig 3A. As we move across the diagram, the degree of alignment of the fish motion—the polarisation—also changes, indicating that changes in the local density (as measured by $\langle l_{\mathrm{nn}} \rangle$) and in the pattern and velocity of motion (as measured by $\langle l_p \rangle$) are reflected in the polarisation of the shoals. For high $\langle l_p \rangle$ and low $\langle l_{\mathrm{nn}} \rangle$, the movements of the fish are cohesive and ordered (S1 Video). For the fish states with a low

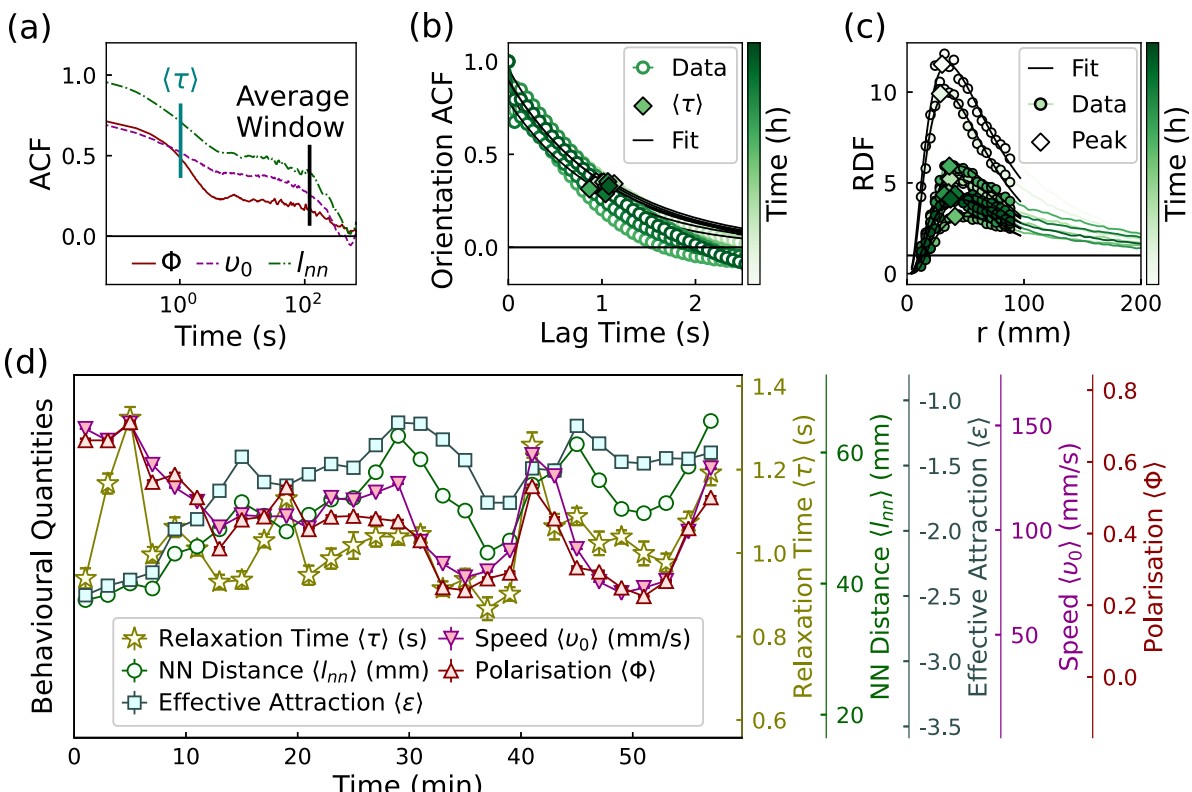

**Fig 2. The behavioural quantities of 50 young zebrafish (group Y1).** A: The auto–correlation function of the polarisation and average speed of the fish group. B: The auto–correlation function of the orientations of fish. C: Sequence of radial distribution functions with increasing time: at early times (top curves) the fish are clustered together so that the peak is large; at later times (bottom curves) the local density decreases and so does the peak height. D: The time evolution of the averaged behavioural quantities for 50 young fish. Each point corresponds to the average value in 2 minutes. The error bars illustrate the standard error values.

$\langle l_{nn} \rangle$ and low $\langle l_p \rangle$, the movements are cohesive but disordered (S3 Video). For fish states with high $\langle l_{nn} \rangle$ and low $\langle l_p \rangle$, the fish are spatially separated with disordered movements (S2 Video). Separated and ordered states are never observed. We also note that there is a systematic difference between young (Y) and old (O) fish groups, with the former characterised by longer persistence lengths, shorter neighbour distances and larger polarisations, while the latter are clustered in a narrower range of persistence lengths with more disorder.

**Table 1. A summary of the variables used to describe the fish behaviour.**

| Symbol | Name | Unit | Comment |
|---|---|---|---|
| $v_0$ | Speed | mm/s | Average over different fish |
| $l_{nn}$ | Nearest Neighbour Distance | mm | Average over different fish |
| $\Phi$ | Polarisation | 1 | Larger = ordered movement |
| $\langle \cdot \rangle$ | Average operator | | Time average over 120 s |
| $\langle \tau \rangle$ | Relaxation time | s | The relaxation of fish orientation |
| $\langle \epsilon \rangle$ | Effective Attraction | 1 | Smaller = more cohesive |
| $\langle l_p \rangle$ | Persistence Length | mm | Defined as $\langle v_0 \rangle \langle \tau \rangle$ |
| $\kappa$ | Reduced Persistence Length | 1 | Defined as $\langle l_p \rangle / \langle l_{nn} \rangle$ |

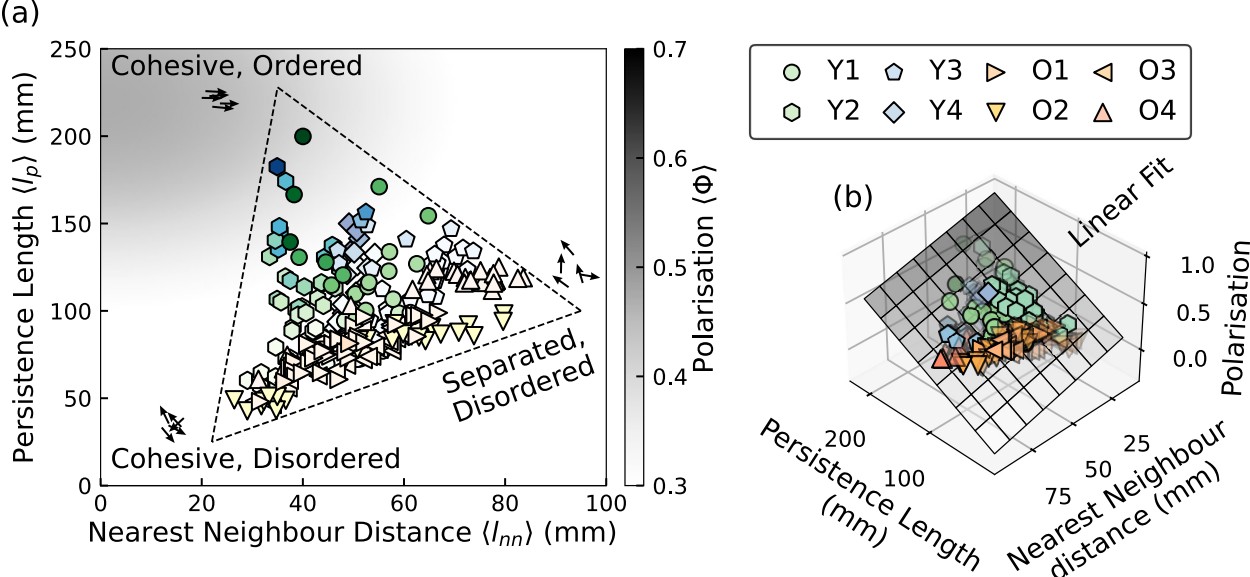

**Fig 3. The states of Zebrafish characterised by two length scales.** A: The states of the fish represented by the nearest neighbour distance and the persistence length. The brightness of the markers corresponds to the value of the polarisation. Each scatter–point corresponds to a time-average of 2 minutes. Different shapes indicate different fish groups from different experiments. B: A multilinear regression model fitting the relationship between the polarisation and the two length scales, indicating the polarisation increase with the increase of persistence length, and the decrease of the nearest neighbour distance. The model is rendered as a 2D plane, whose darkness indicates the value of polarisation.

The simplest model to capture the relationship between polarisation and the two lengths-cales is a multilinear regression. This yields $\langle\Phi\rangle = 0.039\,\langle l_p\rangle - 0.05\,\langle l_{nn}\rangle + 0.147$, with a goodness of fit value $R^2 = 0.73$. This strong simplification suggests that most of the fish macroscopic states reside on a planar manifold in the $\Phi$–$l_{nn}$–$l_p$ space, illustrated in Fig 3B. The value of $\langle\Phi\rangle$ increases with the increase of $\langle l_p\rangle$, and the decrease of $\langle l_{nn}\rangle$. Such relationship is reminiscent of results from the agent-based Vicsek model, where the polarisation of self–propelled particles is determined by the density ($\sim l_{nn}^{-1}$) and orientational noise ($\sim l_p^{-1}$) [21, 45]. In addition, the relationship between the polarisation and the local density suggests a metric based interaction

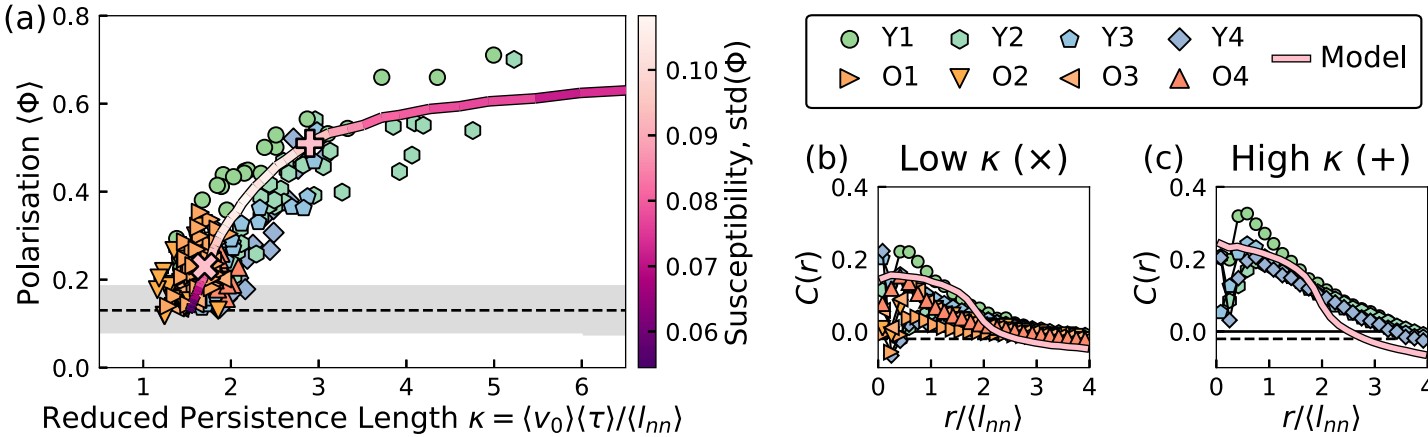

**Fig 4. Single–parameter description of the Zebrafish system.** A: The average polarisation $\langle\Phi\rangle$ as a function of the reduced persistence length $\kappa$, where most data points collapse, and agree with the simulation result of the inertial Vicsek model. The dashed line and grayscale zone represent the expected average value and standard deviation of $\langle\Phi\rangle$ for the uniform random sampling of vectors on the unit sphere. B: The velocity correlation function of the fish and the model in the low $\kappa$ states, highlighted in A with a $\times$ symbol. C: The velocity correlation function of the fish and the model in the high $\kappa$ states, highlighted in A with a + symbol.

rule, rather than the topological one [46]. In other words, the fish tend to align with nearby neighbours, rather than a fixed number of neighbours. For instance, if the fish always align with their closest neighbours regardless of the distance, then the polarisation of the system will not be affected by $l_{nn}$. A similar relationship between polarisation and density was also found for jackdaw flocks while responding to predators [47].

Interestingly, the ratio between the persistence length and the nearest neighbour distance exhibits a simple and robust correlation with the polarisation. Here we introduce the *reduced* persistence length $\kappa = \langle l_p \rangle / \langle l_{nn} \rangle$. The value of $\kappa$ exhibits a consistent relationship with the polarisation for all the fish groups, as shown in Fig 4A. All the experimental data points collapse onto a single curve, especially for the younger fish groups (Y1–Y4) which have a much wider dynamic range than the older groups. Notably, the young fish always transform from ordered states with high $\kappa$ value to disordered states with low $\kappa$ value, possibly because they adapt to the observation tank. Other possible ways to collapse the data is discussed in section VII in S1 Text.

To understand this relationship, we consider the fish motion as a sequence of persistent paths interrupted by reorientations. In a simplified picture, the new swimming direction at a reorientation event is determined by an effective local alignment interaction that depends on the neighbourhood, and notably on the nearest neighbour distance $l_{nn}$. The fish states with larger value of $\kappa$ correspond to situations where each individual fish interacts with more neighbours on average, between successive reorientations. The increased neighbour number leads to a more ordered collective behaviour, so that the values of $\kappa$ and $\Phi$ are positively correlated as shown in Fig 4A.

The time-averaged spatial correlation of the velocity fluctuation supports our picture of the local alignment interaction between the fish. Such a correlation function is defined as,

$$C(r) = \frac{\sum_{i=1}^{N} \sum_{j=i+1}^{N} [(\mathbf{v}_i - \bar{\mathbf{v}}) \cdot (\mathbf{v}_j - \bar{\mathbf{v}}) \ \delta(r - r_{ij})]}{\sum_{i=1}^{N} \sum_{j=i+1}^{N} \delta(r - r_{ij})}, \tag{1}$$

where $\mathbf{v}_i$ is the velocity of fish $i$, $\bar{\mathbf{v}}$ is the average velocity in one frame, $r_{ij}$ is the distance between two fish, and $\delta$ is the Dirac delta function. This function is widely used to characterise the average alignment of velocity fluctuations of moving animals, at different distances [2, 48, 49]. Fig 4B and 4C show the correlation functions for different fish groups with low and high $\kappa$ values, respectively. The distances are rescaled by the different $\langle l_{nn} \rangle$ values of each fish group. For both conditions, the correlation curve collapses beyond one $\langle l_{nn} \rangle$, and peaks around the value of $\langle l_{nn} \rangle$, supporting our assumption that $\langle l_{nn} \rangle$ is the length scale for the fish–fish interactions.

## Vicsek model rationalisation of the experiments

The relationship between $\kappa$ and $\Phi$, presented in Fig 4A, can be easily compared with simulations. Here we explore this through simulations proposing a new modification of the original Vicsek model [21]. The Vicsek model treats the fish as point-like agents with an associated velocity vector of constant speed $v_0$. During the movement, the fish adjust their orientations to align with the neighbours' average moving direction. In order to take into account of memory effects in a simple fashion, we add an inertia term into the Vicsek model, so that each agent partially retain their velocities after the update, with the following rule

$$\mathbf{v}_i(t+1) = v_0 \Theta \left[ (1-\alpha) \underbrace{v_0 \mathcal{R}_\eta \left[ \Theta \left( \sum_{j \in S_i} \mathbf{v}_j(t) \right) \right]}_{\text{Vicsek Model}} + \alpha \mathbf{v}_i(t) \right], \tag{2}$$

where $\vec{v}_i$ is the velocity or the $i$th fish, and the updated velocity of fish $i$ is a linear mixture of its previous velocity and a Vicsek term. The parameter $\alpha$ characterises the proportion of the non-updated velocity, $i.e.$ the inertia. This model is reduced to the Vicsek model by setting $\alpha$ to 0. If $\alpha = 1$, these agents will perform straight motion with constant speed without any interaction. For the Vicsek term, $S_i$ is the set of the neighbours of fish $i$, and the $\Theta$ is a normalising function. The operator $\mathcal{R}_\eta[\mathbf{r}]$ randomly rotates the vector $\mathbf{r}$ into a new direction, which is drawn uniformly from a cap on the unit sphere. The cap is centred around $\mathbf{r}$ with an area of $4\pi\eta$. The value of $\eta$ determines the degree of stochasticity of the system. Our model is thus an inertial Vicsek model in three dimensions with scalar noise.

We set the units of the interaction range $\xi$ and time $dt$ and fix the number density to $\rho = 1\xi^{-3}$ and speed $v = 0.1\xi/dt$. We then proceed with varying the two parameters $\alpha$ and $\eta$ to match the data. In particular we measure the average persistence length $\langle k \rangle$ and polarisation $\langle \Phi \rangle$ and find that for $\alpha = 0.63$ we can fit the data only through the variation of the noise strength $\eta$ (more details of the simulation are available in the section IV in S1 Text). For $\eta \sim 1$, the movement of each agent is completely disordered, reproducing the low $\kappa$ (or $\Phi$) states of the fish. For the case of $\eta \sim 0.65$ the movements of the agents are ordered ($\Phi \sim 0.64$) and mimic the states of fish with high $\kappa$. This is consistent with the fact that in the ordinary Vicsek model the persistence length scales as $\ell \sim v_0/\eta^2$ (section V in S1 Text) [45]. The good fit of the simulation result suggests the fish–fish alignment interaction dominates their behaviour, and the fish can change their states by changing the rotational noise ($\eta$).

We emphasise that the inertial Vicsek model is a crude approximation, as the only interaction of the model is velocity alignment. Without the attractive/repulsive interactions and other details, the inertial Vicsek model does not reproduce the velocity correlation function of the fish, as illustrated in Fig 4B and 4C, suggesting that more sophisticated models with effective pairwise and higher order interactions may be developed in the future. Nevertheless, the model qualitatively reproduces the fact that the velocity correlation is stronger in the high $\kappa$ states.

## Discussion and conclusion

Our results confirm some previous observations and open novel research directions. The young fish appear to adapt to a new environment with the reduction of the effective attraction and speed (Fig 2). Such behaviour is consistent with previous observations of dense groups of fish dispersing over 2–3 hours [50], and it might be related to the fact that the fish perceived less danger as they adapted to the new environment [51, 52]. At the same time, it was reported by Miller and Gerlai that the habituation time has no influence on the Zebrafish group density [53]. We speculate that this difference emerges from the way the statistics were performed. Typically, Miller and Gerlai's results were averaged over 8 different small fish groups (N = 16), and it is possible that the noise among different groups obfuscates the time dependence features here highlighted. Despite the different claims, our result matched Miller and Gerlai's result quantitatively (section VI in S1 Text).

The strong correlation between the speed and polarisation (Fig 4D) is consistent with previous studies on zebrafish [54, 55]. Such correlation had also been observed for different fish species [15, 28, 56]. The polarisation of the young fish was found to decrease (Fig 2D), during the adaptation process. This "schooling to shoaling" phenomenon had also been observed previously in a quasi 2 dimensional environment [51]. Our results, being quantitatively consistent with previous reports (section VI in S1 Text), suggest that this behaviour is present also in a fully three-dimensional context and that the change from schooling (ordered motion) to

shoaling (disordered motion) is related to an increasingly disordered or uncorrelated behaviour, corresponding to the increase in the noise term $\eta$ in the Vicsek model.

It is been speculated that all the biological systems were poised near the critical state, enjoying the maximum response to the environmental stimuli [57]. Here the inertial Vicsek model offers a supporting evidence to this claim. The fluctuation of the polarisation, the susceptibility, took a maximum value at moderate reduced persistence value $\kappa \sim 2$, as illustrated Fig 4A. Also, the fish states were clustered around such region, where the fish can switch between the disordered behaviour and ordered behaviour swiftly. Such disordered but critical behaviour was also observed for the midges in the urban parks of Rome [18].

In conclusion, our work presents a quantitative study of the spatial and temporal correlations manifested by a large group of zebrafish. In our fully 3D characterisation, we have shown that there is a timescale separation between rapid reorientations at short times and the formation of a dynamical state with characteristic spatial correlations at longer times. Such spatial correlations evolve continuously and no steady state is observed in the time window of one hour. Our analysis shows that the continuously changing collective macroscopic states of the fish can be described quantitatively by the persistence length and nearest neighbour distance. The ratio of these length scales presents a characteristic correlation with the polarisation of the fish group. This simple relation is supported by an elementary agent based model in the class of the Vicsek models for collective behaviour.

Our analysis also opens multiple questions: the true nature of the interactions and how these are linked to the sensory and vision capabilities of the fish is open to debate; also, the reason for the change of the fish states remained unexplored, with the possibility of the fish learning over time about the experimental conditions. Our work shows that zebrafish provides a viable model system for the study of animal collective behaviour where such questions can be investigated in a quantitative manner.

A further intriguing possibility is to link the methodology that we develop here, with genetic modifications to zebrafish, for example with behavioural phenomena such as autism [27] or physical alterations such as the stiffened bone and cartilage [58].

## Supporting information

**S1 Text. The details on the experiments, the analysis, and simulations. And the comparison between our results and previous studies.** The detailed description of the tracking system was introduced in the first section, including the analysis on the overall tracking accuracy. The second section presented the details of the fish age. The third section discussed the details of the analysis, typically the calculation of the correlation functions. The fourth section showed the simulation parameters. The fifth section verified the scaling relationship between persistence length and noise in the Vicsek model. The sixth section presented the comparison between our result and previous results. The last section discussed different ways to collapse the experimental data.
(PDF)

**S1 Dataset. The behavioural quantities, correlation functions, and simulation results to produce figures in the main text.** The results were organised in the json file format, and they can be parsed by common json libraries in different computer languages. The pedagogical code (S1 Code) was provided to aid the reader visualising the results.
(JSON)

**S1 Code. The pedagogical code to explore the dataset and to generate the figures in the main text.** The code was written in Python language, and organised as a jupyter notebook. (IPYNB)

**S1 Video. Fifty young fish in the fast and ordered state.** The movie shows the 2D video and 3D trajectories of 50 young fish in real time. In the 3D plot, The dots represent the location of the fish, and the arrows represent the velocities of the fish. The length of the arrow indicates the value of the speed. (MP4)

**S2 Video. Fifty young fish in the slow and disordered state.** The movie shows the 2D video and 3D trajectories of 50 young fish in real time. In the 3D plot, The dots represent the location of the fish, and the arrows represent the velocities of the fish. The length of the arrow indicates the value of the speed. (MP4)

**S3 Video. Fifty old fish in the fast and disordered state.** The movie shows the 2D video and 3D trajectories of 50 old fish in real time. In the 3D plot, The dots represent the location of the fish, and the arrows represent the velocities of the fish. The length of the arrow indicates the value of the speed. (MP4)

**S4 Video. The reprojection error of the 3D tracking of 50 young fish.** The circles are 2D tracking results, and the cross markers are the reprojected 3D tracking results. (MP4)

**S5 Video. The reprojection error of the 3D tracking of 50 simulated fish.** The circles are 2D tracking results, and the cross markers are the reprojected 3D tracking results. (MP4)

## Acknowledgments

We thank James G. Puckett, Christos C. Ioannou, and James Herbert for stimulating conversations. This work was carried out using the computational facilities of the Advanced Computing Research Centre, University of Bristol—http://www.bris.ac.uk/acrc/.

## Author Contributions

**Conceptualization:** Yushi Yang, Francesco Turci, Erika Kague, Chrissy L. Hammond, John Russo, C. Patrick Royall.

**Data curation:** Yushi Yang.

**Formal analysis:** Yushi Yang, Francesco Turci, C. Patrick Royall.

**Investigation:** Yushi Yang.

**Methodology:** Yushi Yang, Francesco Turci, Erika Kague, Chrissy L. Hammond.

**Resources:** Chrissy L. Hammond.

**Software:** Yushi Yang, Francesco Turci.

**Supervision:** John Russo, C. Patrick Royall.

**Validation:** Erika Kague.

**Visualization:** Yushi Yang.

**Writing – original draft:** Yushi Yang, Francesco Turci, Erika Kague, Chrissy L. Hammond, John Russo, C. Patrick Royall.

**Writing – review & editing:** Yushi Yang, Francesco Turci, C. Patrick Royall.

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
