## [Decision Letter · Decision Letter 0]

25 Oct 2021

Dear Mr. Yang,

Thank you very much for submitting your manuscript "Dominating Lengthscales of Zebrafish Collective Behaviour" for consideration at PLOS Computational Biology. As with all papers reviewed by the journal, your manuscript was reviewed by members of the editorial board and by several independent reviewers. The reviewers appreciated the attention to an important topic. Based on the reviews, we are likely to accept this manuscript for publication, providing that you modify the manuscript according to the review recommendations.

Sincerely,

James A.R. Marshall, BSc, PhD

Associate Editor

PLOS Computational Biology

Ville Mustonen

Deputy Editor

PLOS Computational Biology

[LINK]

Reviewer's Responses to Questions

**Comments to the Authors:**

Reviewer #1: This is a review of the manuscript “Dominating Lengthscales of Zebrafish Collective Behaviour”.

The authors present research that examines how various behavior metric depend on two length scales using 3D tracking of schools of 50 zebrafish. The experiments showed that the two length scales – the persistence length and nearest neighbor distance – correlate with the polarization. The authors use simulations to support their claim. This code is made publically available on github – though this could benefit from better readme files and documentation.

In my opinion this work is well-suited for publication in PLOS computational biology given it meets the criteria for publication (originality, innovation, importance to researchers in the field, insight, methodology and conclusions).

I have a few questions and comments that I would like the authors to address/answer:

1) How does the inaccuracy in tracking (Fig S3), where about 90% of fish are tracked each frame, affect the nearest neighbor statistics? It seems to me that tracking is more likely to lose fish that are occluded or near other fish and this may be a systematic error in the nearest neighbor distance metric. This effect even is shown in the figure. How does this impact the ratio and its correlation to polarization?

2) How does the nearest neighbor distance compare with the average distance to neighbors (e.g. through Delauney triangulation)? This may be a more robust statistic.

3) In the SI, you cite an early literature which remarks on the difficulties of stereomatching through an index change (air-water). Since this is not been done in more recent literature, could the authors provide more details about how the index change affects the measurement of the depth of the fish in the tank?

4) Fig 2 is very difficult to read as markers are nearly identical to my eye (diamonds and circles) One solution may be replace the diamonds with stars.

5) Also, in Fig 2, there is a large jump in several quantities at 5 min and 40 min. Could these jumps be ‘startle’ responses? It’s been reported that vibrations (from something like a door closing in another room of the building) can cause the fish to respond like this.

6) What are the uncertainties for the fit to the plan for Fig 3b? How ‘thick’ is this plane?

7) The cameras are reported as Basler acA2040 um, which from a quick search is 3.2MP. Could this information and the size of fish in px be provided in the SI or text?

8) Activity in fish schools has been shown to depend on temperature. The authors state that the fish are kept in water maintained at 25C. In the paper, this is reported as the living tank temperature. I’ll assume that this is the same for the experimental tank. Since this temperature is higher than room temperature, how is the experimental tank maintained at 25C?

9) In the text page 13, the authors report that fish are keep in living tanks with a density of 10fish/L. But some details are missing. Are the same group of fish used for each test (Y1,Y2,O1,O2, etc)? Or are several living tanks collected to make a school of 50 fish for the experiment? How many fish are there in each living tank?

10) Several typos are present in the text an supplemental information. I will include a few here:

Text p8 ‘Tis’ -> this

SI p 6 ‘wold’ -> old

Reviewer #2: Overall, a nice interesting paper. It would benefit from some more controls to be convincing. Regarding language, there is room for improvement, particularly in the methods.

Important comments:

- You put a lot of stress into the fact that the group is studied in 3D and not in 2D. However, Fig 1c suggests that zebrafish are most of the time in a pseudo-2D conformation. Maybe you can report the standard deviation of z and x-y fish locations from the group center of mass?

- I miss some exploration about how other variables fall into the framework of the two length scales. For example, speed seems to correlate a lot with polarisation (Miller&Gerlai and your figure 2), and it seems to increase alignment (e.g. Heras et al. "Deep attention networks reveal the rules of collective motion in zebrafish", 2019), so I suspect you will see some correlation between speed and the length scales. One of the main contributions of the manuscript is to show the two length scales as useful to describe collective motion, so some effort to try to test whether other variables are useful as well (or more useful) is important.

- One of the main results is the relation between \\kappa and the polarisation. Again, is it much better than the relation between speed (or other variable) and polarisation? This is of utmost important, because the Vicsek model you use to validate your ideas has a constant speed for all agents.

- Regarding the discussion, stress/relaxation is mentioned in Miller & Gerlai (2012) as a possible explanation of the decrease in polarization of the group after spending some time in the same tank. There is a nice discussion about this effect in Miller & Gerlai, and also some more modern theoretical studies that are worth mentioning (e.g. Perez-Escudero et al. "Adversity magnifies the importance ofsocial information in decision-making", 2017).

Minor comments, typos and optional suggestions:

A lot of variables to remember, so I think a table of symbols could be useful. Some optional suggestions:

- Maybe all length constants as l with varying subindices: l_0, l_p ...

- Maybe all time constants as \\tau with varying subindices

When possible, I would suggest avoiding the use of several terms for the same thing (confusing to the reader):

- zebrafish/Zebrafish

- group/cluster

- ordered/polarised/alligned

- disordered/randomised

- orientation/direction

pag 2

- is it really "tens of kilometers"?

- The sentence "The emergence of..." contains two parts with two unrelated ideas. What about separating in two sentences.

pag 3

- "birds of shoal" > "birds or shoal"

pag 4

- "and that the density distribution is inhomogenous" is redundant (you say how it is inhomogenous just before)

pag 5

- The polarisation order parameter not exactly as in [20] (in 20 all animals have same speed). I would recommend citing any paper with the same definition (e.g. Cavagna et al.).

pag 6

- "We note that while... ...on time differences". Unsure about what you mean here, or why this is worth noting. Is it a statement about which quantities change if we change the time units?

pag 7

- "we focus on the fish center of mass". Of the group or each of individual fish? I guess the latter, but then unclear why this is mentioned here, as all previous analysis consider fish as point particles.

- "characteristic of Zebrafish behaviour". Maybe remove as it is explained two sentences later? Alternatively, cite 37 here?

pag 8

- "Tis" > "This"

- "polar order" > "polarisation order"

- "Cohesive but dilute". What is dilute? Maybe you mean "Separated and ordered"

- "larger persistence lengths, neighbour distances and polarisations" > "longer persistence lengths, shorter neighbour distances and larger polarisations"

pag 9

- in third line you refer directly to R2 but it is inside brackets

- "... suggests a metric based interaction rule ..." this seems an interesting observation. Maybe you cound give more details about it?

- The sentence "A further simplification..." needs to be rewritten

pag 10

- the definition of C(r) is missing something (what happens to index j ?).

- "constant ballistic movements", maybe better "straight movement with constant speed"?

pag 11

- "the persistence length scales as..." Is this a new result, or something known about the Vicsek model? Needs clarification or citation

pag 12

- Is the distribution of polarisation bimodal, as in Miller&Gerlai 2012?

pag 13

- Missing github links

**Have the authors made all data and (if applicable) computational code underlying the findings in their manuscript fully available?**

Reviewer #1: Yes

Reviewer #2: **No: **I could not check, as missing links in main manuscript

PLOS authors have the option to publish the peer review history of their article (what does this mean?). If published, this will include your full peer review and any attached files.

Reviewer #1: No

Reviewer #2: No

Figure Files:

Data Requirements:

Reproducibility:

References:

---

## [Decision Letter · Decision Letter 1]

9 Dec 2021

Dear Mr. Yang,

We are pleased to inform you that your manuscript 'Dominating Lengthscales of Zebrafish Collective Behaviour' has been provisionally accepted for publication in PLOS Computational Biology.

Best regards,

James A.R. Marshall, BSc, PhD

Associate Editor

PLOS Computational Biology

Ville Mustonen

Deputy Editor

PLOS Computational Biology

Reviewer's Responses to Questions

**Comments to the Authors:**

Reviewer #1: The authors have satisfied all questions and comments. The amendments and edits to the text greatly helped to clarify the work. I recommend that this be published without need for further revision.

Reviewer #2: A suggestion regarding figure R8: I think it is of general interest, so I would include it in the supplement; maybe together with some sort of quantification of the quality of the collapse, if that is possible.

Otherwise, I am fully satisfied with the authors' changes and explanations. Congratulations on the interesting results!

**Have the authors made all data and (if applicable) computational code underlying the findings in their manuscript fully available?**

Reviewer #1: Yes

Reviewer #2: Yes

PLOS authors have the option to publish the peer review history of their article (what does this mean?). If published, this will include your full peer review and any attached files.

Reviewer #1: No

Reviewer #2: No

---

## [Editor Report · Acceptance letter]

7 Jan 2022

PCOMPBIOL-D-21-01528R1 

Dominating Lengthscales of Zebrafish Collective Behaviour

Dear Dr Yang,

I am pleased to inform you that your manuscript has been formally accepted for publication in PLOS Computational Biology. Your manuscript is now with our production department and you will be notified of the publication date in due course.

With kind regards,

Zsanett Szabo
